

# Phylogenetic factorization of compositional data yields lineage-level associations in microbiome datasets

Alex D. Washburne[1], Justin D. Silverman[2,3,4,5], Jonathan W. Leff[6], Dominic J. Bennett[7,8], John L. Darcy[9], Sayan Mukherjee[2,10], Noah Fierer[6] and Lawrence A. David[2,4,5]

[1] Nicholas School of the Environment, Duke University, Durham, NC, United States
[2] Program for Computational Biology and Bioinformatics, Duke University, Durham, NC, United States
[3] Medical Scientist Training Program, Duke University, Durham, NC, United States
[4] Center for Genomic and Computational Biology, Duke University, Durham, NC, United States
[5] Department of Molecular Genetics and Microbiology, Duke University, Durham, NC, United States
[6] Cooperative Institute for Research in Environmental Sciences, University of Colorado, Boulder, CO, United States
[7] Department of Earth Science and Engineering, Imperial College London, London, United Kingdom
[8] Institute of Zoology, Zoological Society of London, London, United Kingdom
[9] Department of Ecology and Evolution, University of Colorado Boulder, Boulder, CO, United States
[10] Department of Statistical Science, Mathematics, and Computer Science, Duke University, Durham, NC, United States

Corresponding author
Alex D. Washburne,
alex.d.washburne@gmail.com

## ABSTRACT

Marker gene sequencing of microbial communities has generated big datasets of microbial relative abundances varying across environmental conditions, sample sites and treatments. These data often come with putative phylogenies, providing unique opportunities to investigate how shared evolutionary history affects microbial abundance patterns. Here, we present a method to identify the phylogenetic factors driving patterns in microbial community composition. We use the method, "phylofactorization," to reanalyze datasets from the human body and soil microbial communities, demonstrating how phylofactorization is a dimensionality-reducing tool, an ordination-visualization tool, and an inferential tool for identifying edges in the phylogeny along which putative functional ecological traits may have arisen.

# INTRODUCTION

Microbial communities play important roles in human (*Human Microbiome Project Consortium, 2012*), livestock (*Gregg, 1995*) and plant (*Berendsen, Pieterse & Bakker, 2012*) health, biogeochemical cycles (*Bardgett, Freeman & Ostle, 2008*; *Falkowski, Fenchel & Delong, 2008*), the maintenance of ecosystem productivity (*Van Der Heijden, Bardgett & Van Straalen, 2008*), and bioremediation (*Li et al., 2000*). Given the importance of microbial communities and the vast number of uncultured and undescribed microbes associated with animal and plant hosts and in natural and engineered systems,

PeerJ ____________________________________________

understanding the factors determining microbial community structure and function is major challenge for modern biology.

Marker gene sequencing (e.g., 16S rRNA gene sequencing to assess bacterial and archaeal diversity and 18S markers for Eukaryotic diversity) is now one of the most commonly used approaches for describing microbial communities, quantifying the relative abundances of individual microbial taxa, and characterizing how microbial communities change across space, time, or in response to known biotic or abiotic gradients.

Analyzing these data is challenging due to the peculiar noise structure of sequence-count data (*Robinson, McCarthy & Smyth, 2010*), the inherently compositional nature of the data (*Friedman & Alm, 2012*), deciding the taxonomic scale of investigation (*Cracraft, 1983*; *Cracraft, 2000*; *Tikhonov, Leach & Wingreen, 2015*), and the high-dimensionality of species-rich microbial communities (*Fierer & Jackson, 2006*). There is a great need and opportunity to develop tools to more efficiently analyze these datasets and leverage information on the phylogenetic relationships among taxa to better identify which clades are driving differences in microbial community composition across sample categories or measured biotic or abiotic gradients (*Martiny et al., 2015*).

Many of these challenges can be resolved by performing regression on clades identified in the phylogeny. In this paper, we take on these challenges by developing a means to perform regression of biotic/abiotic gradients on variables corresponding to branches in the phylogenetic tree, thereby allowing dimensionality reduction with a clear phylogenetic interpretation and consistent with the compositional nature of the data.

Consider a study on the effect of oxazolidinones, which affect gram-positive bacteria, on microbial community composition. Rather than regression of antibiotic treatment on abundance at numerous taxonomic levels, statistical analysis of bacterial communities treated with an oxazolidinone should instantly identify the split between gram-positive and gram-negative bacteria as the most important phylogenetic factor determining response to oxazolidinones. Subsequent factors should then be identified by comparing bacteria within the previously-identified groups: identify clades within gram-positives which may be more resistant or susceptible than the remaining gram-positives. Splitting the phylogeny at each inference and making comparisons within the split groups ensures that subsequent inferences are independent of the gram positive–gram negative split which we have already obtained. All of this analysis must be done consistent with the compositional nature of sequence count data.

Here, we provide a method to analyze phylogenetically-structured compositional data. The algorithm, referred to as "phylofactorization," iteratively identifies the most important clades driving variation in the data through their associations with independent variables. Clades are chosen based on some metric of the strength or importance of their regressions with meta-data, and subsequent clades are chosen by comparison of sub-clades within the previously-identified bins of phylogenetic groups. Each "factor" identified corresponds to an edge in the phylogeny, and phylofactorization builds on literature from compositional data analysis to construct a set of orthogonal axes corresponding to those edges; the output orthonormal basis allows the projection of sequence-count relative abundances onto these phylogenetic axes for dimensionality reduction, visualization, and standard multivariate
statistical analyses. The visualizations and inferences drawn from phylofactorization can be tied back to splits in a given phylogenetic tree and thereby allow researchers to annotate the microbial phylogeny from the results of microbiome datasets.

We show with simulations that phylofactor is able to correctly identify affected clades. We then phylofactor a dataset of human oral and fecal microbiomes to determine the phylogenetic factors driving variation in human body site (*Caporaso et al., 2011*), and a dataset of soil microbes using a multiple regression of pH, carbon concentration and nitrogen concentration (*Ramirez et al., 2014*). In the human microbiome dataset, we find three splits in the phylogeny that together capture 17.6% of the variation community composition across two body sites. We use phylofactorization to find that the dominant features driving body-site variation are invisible to taxonomy-based analyses, including splits between unclassified OTUs, monophyletic clades that span several taxonomic groups, and a spectrum of phylogenetic scales for binning OTUs based on habitat preferences. In the soil microbiome dataset, we use phylofactor-based dimensionality reduction to visualize and quantitatively confirm that pH drives most of the variation in the soil dataset but we emphasize that the axes from phylofactor ordination-visualization plots correspond to identifiable edges on the phylogeny that have clear biological interpretations and can be used and tested across studies. User-friendly code for implementing, summarizing and visualizing phylofactorization is provided in an R package ('phylofactor'), and a tutorial is available online.

## MATERIALS & METHODS

### Phylogenetically-structured compositional data

Microbiome datasets are "phylogenetically-structured compositional data," compositions of parts linked together by a phylogeny for which only inferences on relative abundances can be drawn. The phylogeny is the scaffolding for the evolution of vertically-transmitted traits, and vertically-transmitted traits may underlie an organism's functional ecology and response to perturbations or environmental gradients. Performing inference on the edges in a phylogeny driving variation in the data can be useful for identifying clades with putative traits causing related taxa to respond similarly to treatments, but such inferences must account for the compositional nature of the sequence-count data.

A standard analysis of microbiome datasets uses only the distal edges of the tree—the OTUs—and a few edges within the tree separating Linnaean taxonomic groups. However, a phylogeny of $D$ taxa and no polytomies is composed of $2D - 3$ edges, each connecting two disjoint sets of taxa in the tree with no guarantee that splits in Linnaean taxonomy corresponds to phylogenetic splits driving variation in our dataset. Thus, instead of analyzing just the tips and a series of Linnaean splits in the tree, a more robust analysis of phylogenetically-structured compositional data should analyze all of the edges in the tree. To do that, we draw on the isometric log-ratio transform from compositional data analysis, which has been used to search for a taxonomic signature of obesity in the human gut flora (*Finucane et al., 2014*) and incorporated into packages for downstream principal components analysis (*Le Cao et al., 2016*). However, to the best of our knowledge, the
previous literature using the isometric log-ratio transform in microbiome datasets has used random or standard sequential binary partitions, and not explicitly incorporated the phylogeny as their sequential binary partition.

## The isometric log-ratio transform of a rooted phylogeny

The isometric log-ratio (ILR) transform was developed as a way to transform compositional data from the simplex into real space where standard statistical tools can be applied (*Egozcue et al., 2003*; *Egozcue & Pawlowsky-Glahn, 2005*). A sequential binary partition is used to construct a new set of coordinates, and the phylogeny is a natural choice for the sequential binary partition in microbiome datasets. Instead of analyzing relative abundances, $y_i$, of $D$ different OTUs, the ILR transform produces $D-1$ coordinates, $x_i^*$ (called "balances"). Each balance corresponds to a single internal node of the tree and represents the averaged difference in relative abundance between the taxa in the two sister clades descending from that node (the difference being appropriately measured as a log-ratio due to the compositional nature of the data; see Supplemental Information 1 for more detailed description of the ILR transform). For an arbitrary node indicating the split of a group, $R$ with $r$ elements from the group, $S$ with $s$ elements, the ILR balance can be written as

$$x_{\{R,S\}}^* = \sqrt{\frac{rs}{r+s}}\log\left(\frac{g(\mathbf{y}_R)}{g(\mathbf{y}_S)}\right) \tag{1}$$

where $g(\mathbf{y}_R)$ is the geometric mean of all $y_i$ for $i \in R$.

We refer to the ILR transform corresponding to a rooted phylogeny as the "rooted ILR". The rooted ILR creates a set of ILR coordinates, $\{x_i^*\}$, where each coordinate corresponds to the "balance" between sister clades at each split in the phylogenetic tree. The balances in a rooted ILR transform in equation Eq. (1) can be intuited as the average difference between taxa in two groups, and splits in the tree which meaningfully differentiate taxa will be those splits in which the average difference between taxa in two groups changes predictably with an independent variable. Inferences on ILR coordinates, then, map to inferences on lineages in the phylogenetic tree.

The rooted ILR coordinates provide a natural way to analyze microbiota data as they measure the difference in the relative abundances of sister clades and may be useful in identifying effects contained within clades such as zero-sum competition of close relatives or the substitution of one relative for another across environments. However, if we desire to link the effect of a design variable or an external covariate (e.g., antibiotics vs. no antibiotic treatment) to clades within the phylogeny, the best comparison may not be between sister clades, but instead between all other clades, controlling for any other phylogenetic splits or factors we may know of (e.g., we may compare a lineage within gram-positives with all other gram-positives, once we've identified the gram-positive vs. gram-negative split as an important factor for antibiotic susceptibility). We refer to this unrooted approach as 'phylofactorization.'

For the task of linking an external covariate to individual clades in the phylogeny, we examine three features of the rooted ILR that can be improved on by phylofactorization by considering a treatment that decreases the abundance of one and only one clade, $B$,

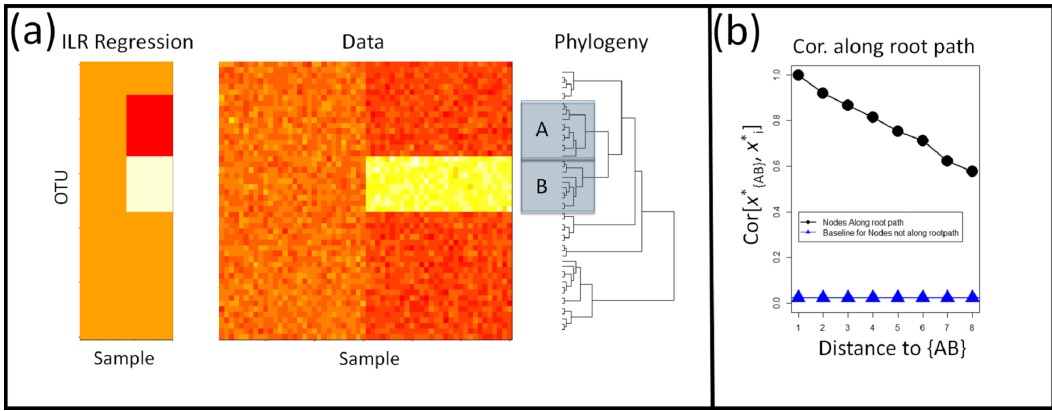

**Figure 1 Shortcomings of Rooted ILR.** (A) The isometric log-ratio transform corresponding to a phylogeny rooted at the common ancestor is inaccurate for geometric changes within clades. Here, absolute abundances of 50 taxa in 30 samples per site were simulated across two sites. An affected clade, $B$, is up-represented in the second site. Regression on the rooted ILR coordinates, $x_i^*$, against the sample site indicated that the partition separating clade $A, B$, referred to as $x_{\{A,B\}}^*$, had the highest test-statistic, but the rooted ILR predicts fold-changes in $B$ relative to $A$, not fold changes in $B$ relative to the rest of the taxa. (B) Consequently, when one clade increase in abundance while the rest remain unaffected, partitions between the affected clade and the root will also have a signal leading to a correlation in the coordinates along the path from $B$ to the root. The correlation plotted here is the absolute value of the correlation coefficient, and the baseline correlation was estimated as the average absolute value of the correlation coefficient between ILR coordinates not along the root-path of the affected clade.

whose closest relative is clade $A$. Regression on the rooted ILR coordinates may identify the balance $x_{\{A,B\}}^*$ corresponding to the most recent common ancestor of clades $A$ and $B$ as having that strongest response to the treatment, but regression on this coordinate will suggest that clade $B$ decreases relative to $A$, leading to structured residuals in the original dataset due to an inability to account for the increase in clade $B$ relative to the rest of the OTUs in the data (Fig. 1A). Second, all partitions between the affected clade and the root will be affected. If each balance is tested independently, the rooted ILR may identify many clades that are affected by antibiotics; the correlations between coordinates can yield a high false-positive rate if just one clade is affected (Fig. 1B). Finally, the ILR transformation does not work with polytomies common in real, unresolved phylogenies. Any polytomy will produce a split in the phylogeny between three or more taxa, and there is no general way to describe the balance of relative abundances of three or more parts using only one coordinate.

Nonetheless, the simplicity and theoretical foundations underlying the ILR, and the instant appeal of applying it to the sequential-binary partition of the phylogeny, motivate the rooted ILR as a simple tool for analysis of the phylogenetic structure in compositional data. For that reason, we use the rooted ILR as a baseline for comparison of our more complicated method of phylogenetic factorization.

## Phylofactorization

The shortcomings of the rooted ILR can be remedied by modifying the ILR transform to apply not to the nodes or splits in a phylogeny, but to the edges in an unrooted phylogeny.
While ILR coordinates of nodes allow a comparison of sister clades, ILR coordinates along edges allow comparison of taxa with putative traits that arose along the edge against all taxa without those putative traits. Traits arise along edges of the phylogeny and so, for annotation of phylogenies, effects in a clade are best mapped to a chain of edges in the phylogeny.

However, the ILR transform requires a sequential binary partition, and the edges don't immediately provide a clear candidate for a sequential binary partition. In what we refer to as "phylofactorization," one can iteratively construct a sequential binary partition from the unrooted phylogeny by using a greedy algorithm by sequentially choosing edges which maximize a researcher's objective function. Phylofactorization consists of 3 steps (Fig. 2): (1) Consider the set of possible primary ILR basis elements corresponding to a partition along any edge in the tree (including the tips). (2) Choose the edge whose corresponding ILR basis element maximizes some objective function—such as the test-statistic from regression or the percent of variation explained in the original dataset—and the groups of taxa split by that edge form the first partition. (3) Repeat steps 1 and 2, constructing subsequent ILR basis elements corresponding to remaining edges in the phylogeny and made orthogonal to all previous partitions by limiting the comparisons to taxa within the groups of taxa un-split by previous partitions.

Explicitly, the first iteration of phylofactorization considers a set of candidate ILR coordinates, $\{x_e^*\}$ corresponding to the two groups of taxa split by each edge, $e$. Then, regression is performed on each of the ILR coordinates, $x_e^* \sim f(X)$ for an appropriate function, $f$ and a set of independent variables, $X$. The edge, $e_1^+$, which maximizes the objective function is chosen as the first phylogenetic factor. In this paper, our objective function is the difference between the null deviance of the ILR coordinate and the deviance of the generalized linear model explaining that ILR coordinate as a function of the independent variables. We use this objective function as a measure of the amount of variance explained by regression on each edge because the total variance in a compositional dataset is constant and equal to the sum of the variances of all ILR coordinates corresponding to any sequential binary partition. Consequently, at each iteration there is a fixed amount of the total variance remaining in the dataset, and so at the candidate ILR coordinate which captures the greatest fraction of the total variance in the dataset is the one with the greatest amount of variance explained by the regression. After identifying $e_1^+$, we cut the tree in two sub-trees along the edge, $e_1^+$.

For the second iteration, another set of candidate ILR coordinates is constructed such that their underlying balancing elements are orthogonal to the first ILR coordinate. Orthogonality is ensured by constructing ILR coordinates contrasting the abundances of taxa along each edge, restricting the contrast to all taxa within the sub-tree in which the edge is found. A new edge, $e_2^+$, which maximizes the objective function is chosen as the second factor, the sub-tree containing this edge is cut along this edge to produce two sub-trees, and the process is repeated until a desired number of factors is reached or until a stopping criterion is met. More details on the algorithm, along with a discussion on objective functions, is contained in Supplemental Information 1.

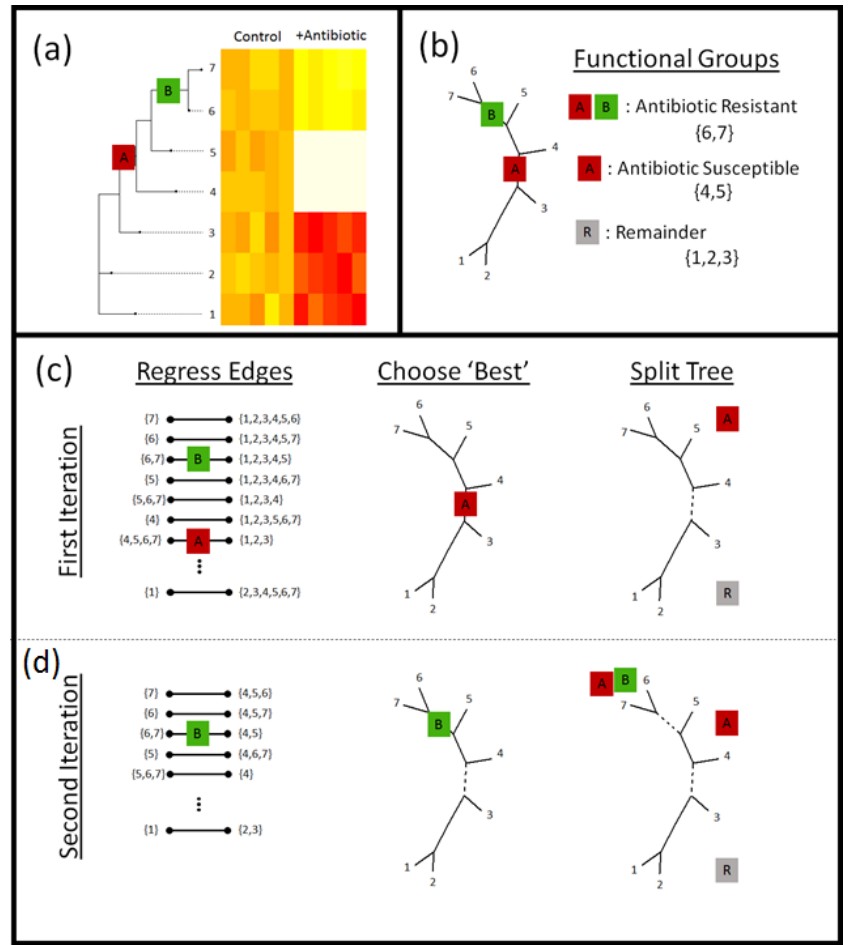

**Figure 2** **Phylofactorization.** (A) Phylofactorization changes variables from tips of the phylogeny (OTUs used in analysis of microbiome datasets) to edges of the phylogeny with the largest predictable differences between taxa on each side of the edge. To illustrate this method, we consider the treatment of a bacterial community with an oxazolidinone. Oxazolidinones target gram-positive bacteria and will likely lead to a decrease in the relative abundances of gram-positive bacteria (antibiotic susceptible clade, *A*, having the antibiotic target). Among the antibiotic susceptible bacteria, phylofactor can identify monophyletic clades that are resistant relative to other antibiotic-susceptible bacteria due to a vertically-transmitted trait (B) such as the loss of the antibiotic target or enzymes that break down the antibiotic. (B) The two phylogenetic factors produce three meaningful bins of taxa—those susceptible to antibiotics (A), those within the susceptible clade that are resistant to antibiotics (A + B), and a potentially paraphyletic remainder. (C) Phylofactorization is a greedy algorithm to extract the edges which capture the most predictable differences in the response of relative abundances among taxa on the two sides of each edge. (C) For the first iteration, all edges are considered—an ILR coordinate is created for each edge using Eq. (1) and the ILR coordinate is regressed against the independent variable. The edge which maximizes the objective function is chosen. Depicted above, the first factor corresponds to the edge separating antibiotic susceptible bacteria from the rest. Then, the tree is split—all subsequent comparisons along edges will be contained within the sub-trees. The conceptual justification for limiting comparisons within sub-trees is to prevent over-lapping comparisons: once we identify the antibiotic susceptible clade, we want to look at which taxa within that clade behave differently from other taxa within that clade. (D) For the second iteration, the remaining edges are considered, ILR coordinates within sub-trees are constructed. The edge maximizing the objective function is selected and the tree is split at that edge. For more details, see the section "PhyloFactor" in Supplemental Information 5.

While one could use other methods of amalgamating abundances along edges, the conceptual importance of using the ILR transform is twofold: the ILR transform has proven asymptotic normality properties for compositional data to allow the application of standard multivariate methods (*Egozcue et al., 2003*), and the ILR transform serves as a measure of contrast between two groups. The log-ratio used in phylofactor is an averaged ratio of abundances of taxa on two sides of an edge (see Supplemental Information 1 for more detail), thus phylofactorization searches the tree for the edge which has the most predictable difference between taxa on each side of the edge or, put differently, the edge which best differentiates taxa on each side. Thus, each edge that differentiates taxa and their responses to independent variables is considered a phylogenetic "factor" driving variation in the data.

The output of phylofactorization is a set of orthogonal, sequentially "less important" ILR basis elements, their predicted balances, and all other information obtained from regression. After the first iteration of phylofactorization, we are left with an ILR basis element corresponding to the edge which maximized our objective function and split the dataset into two disjoint sub-trees, or sets of OTUs that we henceforth refer to as "bins," and we have an estimated ILR balancing element, $\hat{x}_1^*(X)$, where $X$ is our set of independent variables. Subsequent factors will split the bins from previous steps, and after $n$ iterations one has $n$ factors that can be mapped to the phylogeny, $n+1$ bins for binning taxa based on their phylogenetic factors, $n$ estimates of ILR balancing elements, and an orthonormal ILR basis that can be used to project the data onto a lower dimensional space. The sequential splitting of bins in phylofactorization ensures sequentially independent inferences: having already identified group $B$ as hyper-abundant relative to group $A$ in the example illustrated in Fig. 2, downstream factors must analyze sub-compositions entirely within $B$ and within $A$.

## Computational tools

Phylofactorization was done using the R package "phylofactor" available at https://github.com/reptalex/phylofactor. The R package contains detailed help files that demo the use of the package, and the exact code used in analyses and visualization in this paper are available in Supplemental Information 1. The rooted ILR transform was performed as described in *Egozcue & Pawlowsky-Glahn (2005)* where the sequential binary partition was the rooted phylogeny.

## Power analysis of rooted ILR and phylofactorization

To compare the ability of phylofactorization and the rooted ILR to identify clades of OTUs with shared associations with independent variables, we simulated random communities of $D = 50$ OTUs and $p = 40$ samples by simulating random absolute abundances, $N_{i,j}$, such that $\log(N_{i,j})$ were i.i.d Gaussian random variables with mean $\mu = 8$ and standard deviation $\sigma = 0.5$. The OTUs were connected by a random tree (the tree remained constant across all simulations), and then either 1 or 3 clades were randomly chosen to have associations with a binary "environment" independent variable with $p = 20$ samples for each of its two values to represent an equal sampling of microbial communities across two environments.

For simulations with one significant clade, the abundances of all the OTUs within that clade increased by a factor a in the second environment where a $\in \{1.5, 3, 6\}$. For simulations

with three significant clades, the three clades were drawn at random and randomly assigned a fold-change from the set $\{\pi^b, 0.5^b, \exp(-b)\}$ in a randomly chosen environment where $b \in \{1, 2, 5\}$. For each fold-change, 500 replicates were run to compare the power of the rooted ILR and phylofactorization in correctly identifying the affected clades.

Regression of rooted ILR coordinates was performed and the coordinates were ranked by the difference between their null deviance and the model deviance. The ability of a rooted ILR coordinate to identify the correct 1 clade or 3 clades was measured by the percent of its top 1 or 3 ILR coordinates, respectively, which corresponded to the node on the tree from which the affected clade(s) originated. The ability of phylofactor to identify the correct 1 clade or 3 clades was measured by the percent of the factors that correctly split an affected clade from the rest (e.g., the percent of factors corresponding to edges along which a trait arose).

For the 3 clade simulations, we also compared the amount of variance explained by three factors in phylofactorization with the amount of variance explained by the top 3 ILR coordinates in the rooted ILR. The amount of variance explained was measured as the difference in the null deviance and the model deviance, summed across all three factors or the top 3 ILR coordinates.

## KS-based stopping function for phylofactor

While a researcher can iterate through phylofactorization until a full basis of $D-1$ ILR coordinates is constructed, the researcher may be interested in stopping the iteration before the full basis is constructed and focus their analysis and interpretations on a conservative subset of the true number of phylogenetic factors. We implemented a stopping function based on a Kolmogorov–Smirnov (KS) test of the distribution of $P$-values from analyses of variance of the regressions on candidate ILR coordinates. If there is no phylogenetic signal, we anticipate the true distribution of $P$-values to be uniform (albeit with some dependence among the $P$-values due to overlap in the OTUs used in the ILR coordinates). Thus, we tested the ability of phylofactor to correctly identify the number of clades if phylofactorization is stopped when a KS test of the $P$-values produces its own $P$-value $P_{KS} > 0.05$.

We simulated 300 replicate communities with $M$ clades for each $M \in \{1, \ldots, 10\}$. For simulations with $M$ clades, $D = 50$ and $p = 40$ communities were simulated as above and fold changes, $c$, were drawn as log-normal random variables where $\log(c_k)$ were i.i.d Gaussian random variables with $\mu = 0$ and $\sigma = 3$ for $k = 1, \ldots, M$. The number of clades identified by phylofactor for a given true number of clades, $K_{M,r}$, was tallied for $r = 1, \ldots, 300$. We calculate the mean $\bar{K}_M$ across all replicates and, for visualization purposes, interpolate the $\alpha = 0.025$ and $\alpha = 0.975$ quantiles by finding the best fit of a logistic function to the cumulative distribution of $\{K_{M,r}\}_{r=1}^{r=300}$ for each $M$.

## Analysis of fecal/oral microbiome data

16S amplicon sequencing data from *Caporaso et al. (2011)* were downloaded from the MG-RAST database (http://metagenomics.anl.gov/) along with associated metadata. QIIME (*Caporaso et al., 2012*) was used to trim primers from these data, and to cluster OTUs

with the Greengenes reference database (May 2013 version; http://greengenes.lbl.gov). Longer sequence lengths in the greengenes database (~1,400 BP) compared to the original Illumina sequences (~123 BP) allows more informative base pairs for phylogenetic tree construction. We used the phylogenetic tree that is included with the greengenes database for all analyses. The resulting OTU table was rarefied to 6,000 sequences per sample.

A total of 10 time points were randomly drawn from each of the male tongue, female tongue, male feces and female feces datasets, giving a total of $n = 20$ samples at each site. Taxa present in fewer than 30 of the 40 samples were discarded, and phylofactorization was done by adding pseudo-counts of 0.65 to all 0 entries in the dataset (*Aitchison, 1986*), converting counts in each sample to relative abundances, and then regressing the ILR coordinates against body site. The complete R script is available in the file "Data Analysis pipeline of the FT microbiome."

Complete phylofactorization of this dataset was performed by stopping the algorithm when a *KS*-test on the uniformity of *P*-values from analyses of variance of regression on candidate ILR-coordinates yielded $P_{KS} > 0.05$. These results were compared with a standard, multiple hypothesis-testing analysis of CLR-transformed data. The summary of the taxonomic detail at the first three factors is provided in the results section, and a full list of the taxa factored at each step is available in the supplement and can be further explored using the R pipeline provided.

## Analysis of soil microbiome data

The soil microbiome dataset from *Ramirez et al. (2014)* was included to illustrate the ability of phylofactor to work on bigger microbiome datasets with continuous independent variables and multiple regression. Details on sample collection, sequencing, meta-data measurements and OTU clustering are available in *Ramirez et al. (2014)*. The phylogeny was constructed by aligning representative sequences using SINA (*Pruesse, Peplies & Glöckner, 2012*), trimming bases that represented gaps in ≥20% of sequences, and using fasttree (*Price, Dehal & Arkin, 2010*).

The complete dataset contained 123,851 OTUs and 580 samples. Data were filtered to include all OTUs with on average 2 or more sequences counted across all samples, shrinking the dataset to $D = 3,379$ OTUs. The data were further trimmed to include only those samples with available pH, C and N meta-data, reducing the sample size to $n = 551$.

Phylofactorization was done by adding pseudo-counts of 0.65 to all 0 entries in the dataset (*Aitchison, 1986*), converting counts in each sample to relative abundances, and performing multiple regression of pH, C and N on ILR coordinates. The first three factors are used for ordination-visualization. To determine the relative importance of each abiotic variable in driving phylogenetic patterns of microbial community composition, we used the lmg method from the R package 'relaimpo' (*Grömping, 2006*) which averages the sequential sums of squares over all orderings of regressors to obtain a measure of relative importance of each regressor in the multivariate model.

## RESULTS

We find three main results. First, we find that our algorithm out-performs a standard tool for analyzing compositions of parts related by a tree —what we refer to as the "rooted ILR" transform—and that we can obtain a conservative estimate of the number of phylogenetic factors in simulated datasets a with a known number of affected clades. Second, we phylofactor a dataset of the human oral and fecal microbiomes and find that the three dominant edges in the phylogeny account for 17.6% of the variation in microbial communities across these sample sites, edges which are not assigned a unique taxonomic label and are thus difficult or impossible to obtain from taxonomic-based analyses. Third, we show that phylofactorization can be combined with multiple regression to reveal that pH drives the main phylogenetic patterns of community composition in soil microbiomes, and show that in four factors we split the Acidobacteria three times— including one split that identifies a monophyletic clade of Acidobacteria that consists of alkaliphiles. Finally, using the soil dataset, we demonstrate how phylofactorization yields two complimentary methods for dimensionality reduction and ordination-visualization that tell a simplified story of how the major phylogenetic groups of OTUs change with pH. Throughout our results, we emphasize that the phylogenetic inferences made by phylofactorization could be invisible to taxonomic-based analyses, and we conceptually compare the dimensionality-reductions of phylofactorization to the less-interpretable output from standard ordination-visualization tools.

### Power analysis and conservative stopping of phylofactorization

Phylofactorization remedies the structured residuals from the rooted ILR regression on data with fold-changes in abundances within clades. Phylofactorization also remedies the problem of high false-positive rates arising from the nested-dependence and correlated coordinates of the rooted ILR transform, as sequential inferences in phylofactorization are independent. Phylofactorization out-performs the rooted ILR in identifying the correct clades with a given fold-change in abundance (Figs. 3A and 3B), and can be paired with other algorithms assessing residual structure to stop factorization when there is no residual structure and thus accurately identify the number of affected clades (Fig. 3C). Finally, by focusing the inferences on edges instead of nodes in the phylogeny, this algorithm works on trees with polytomies and doesn't require a forced resolution of polytomies to construct a sequential binary partition of the OTUs.

### *Oral-fecal microbiome*

Performing regression of sample site on centered log-ratio (CLR) transformed OTU tables, with 290 OTUs and 40 samples, yielded 236 significant OTUs at a false-discovery rate of 1%; the phylogenetic signal of these OTUs from CLR-based inferences may be difficult to parse out. Phylofactorization of the oral-fecal microbiome dataset yielded 142 factors, the top three of which explain 17.6% of the variation in the dataset, factors which correspond to clearly visible blocks in phylogenetic heatmaps of the OTU table (Fig. 4). The factors span a range of taxonomic scales and all of them would be invisible to taxonomic-based analyses. Below, we summarize the factors—the *P*-values from regression, the taxa split at

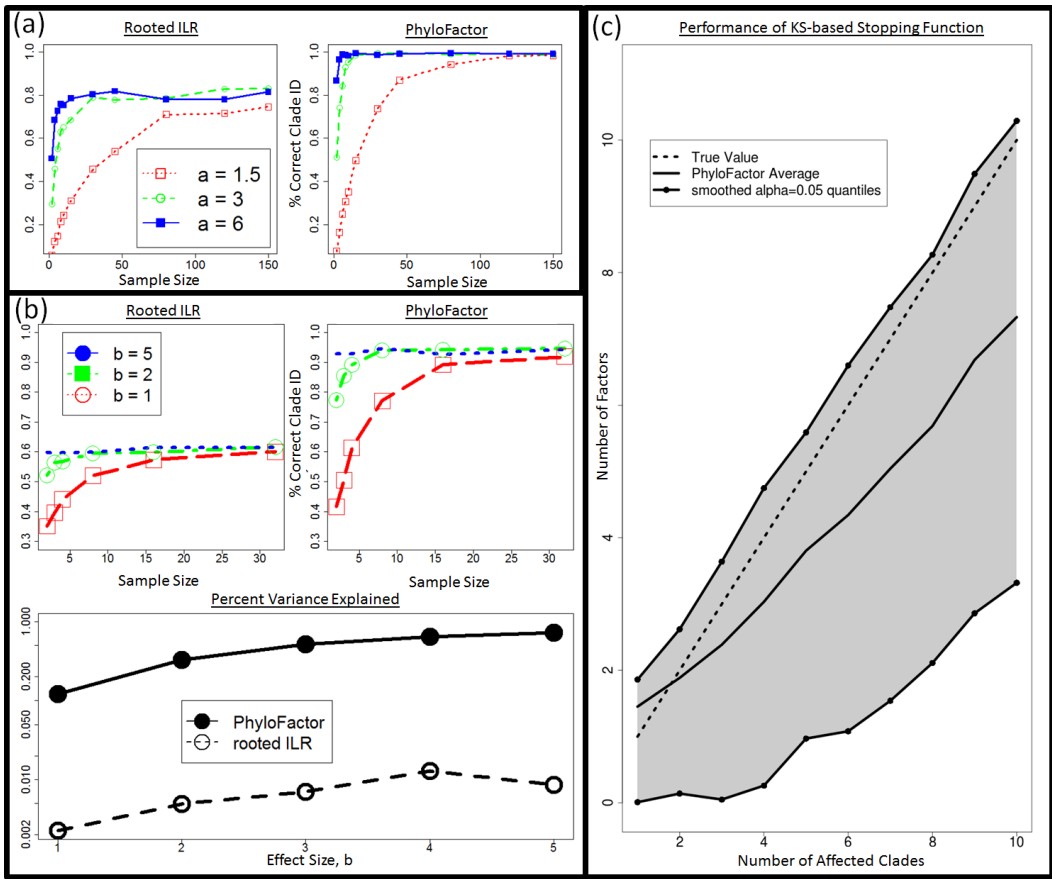

**Figure 3 Phylofactorization can correctly identify affected clades and be stopped at a conservative number of phylogenetic factors.** (A) Power analysis—1 clade. The rooted ILR transform that minimizes residual variance when regressed against sample site is less able to identify the correct clade compared to phylofactorization for a variety of effect sizes, a, and sample sizes. (B) Three Significant Clades: when three significant clades are chosen and given a set of effects increasing in intensity with the parameter b, choosing the top rooted ILR coordinates under performs phylofactorization in correctly identifying the affected clades. Phylofactorization also explains more variation in the data: across effect sizes, phylofactorization explains 2 orders of magnitude more of the variance in the dataset than the sequential rooted ILR. (C) Stopping Phylofactorization: Plots of the true number of affected clades in simulated datasets against the number of clades identified by the R package 'phylofactor.' One can terminate phylofactorization when the true number of affected clade is unknown by choosing a stopping function aimed at stopping when there is no evidence of a remaining signal. By stopping the iteration when the distribution of $P$-values from analyses of variance of regression on candidate ILR basis elements is uniform (specifically, stopping when a KS test against a uniform distribution yields $P > 0.05$), we obtain a conservative estimate of the number of phylogenetic factors in the data.

each factor, the body site associations predicted by generalized linear modeling of the ILR coordinate against body site, and finer detail about the taxonomic identities and known ecology of monophyletic taxa being split. Phylofactorization of these data indicates that a few clades explain a large fraction of the variation in the data, and many more clades can be identified as containing the same intricate detail as the phylogenetic factors presented below. The biology of microbial human-body-site association can focus on these dominant

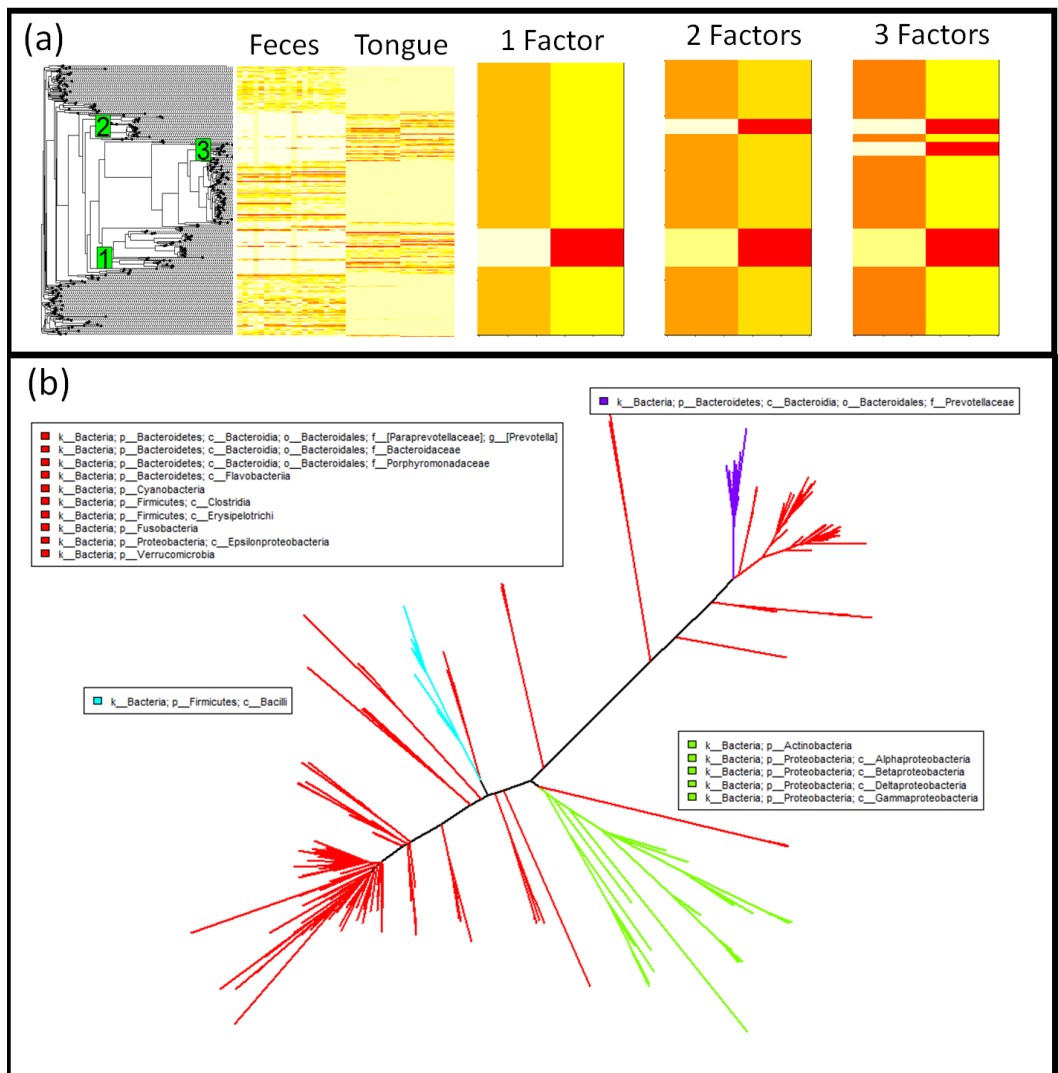

**Figure 4  Phylofactorization of human feces/tongue dataset identifies clades differentiating body sites.**
(A) Phylogenetic structure is visible as blocks using a phylogenetic heatmap from the R package 'phytools' (*Revell, 2012*). The first factor separates Actinobacteria and some Proteobacteria from the rest, the second factor separates the class Bacilli from the remaining non-Proteobacteria and non-Actinobacteria, the third factor pulls out the genus Prevotella from Bacteroidetes and indicates that it, unlike many other taxa in Bacteroidetes, is unrepresented in the tongue. Each factor captures a major block of variation in the data, and the orthogonality of the ILR coordinates from each factor allow multiple factors to be combined easily for estimates of community composition. (B) These three factors splits the phylogeny into four bins. Three of those bins are monophyletic and the final bin is a "remainder" bin, containing taxa split off by the previous monophyletic bins. The three factors are identifiable edges between nodes that can be mapped to an online database containing those nodes. The taxonomic assignment used here is the set of all shortest-unique-prefixes that separate the taxa in each bin.

factors—which traits and evolutionary history drive these monophyletic groups' strong, common association with body sites?

The first factor ($P = 4.90 \times 10^{-30}$) split Actinobacteria and Alpha-, Beta-, Gamma-, and Delta-proteobacteria from Epsilonproteobacteria and the rest (Fig. S4). The underlying generalized linear model predicts the Actinobacteria and non-Epsilon-proteobacteria to be $0.4\times$ as abundant as the rest in the gut and $3.6\times$ as abundant as the rest in the tongue. The Actinobacteria identified as more abundant in the tongue include four members of the plaque-associated family Actinomycetaceae, one unclassified species of *Cornybacterium,* three members of the mouth-associated genus *Rothia* (*Koren et al., 2011*), and one unclassified species of the vaginal-associated genus *Atopobium* (*Ding & Schloss, 2014*). With a standard multivariate analysis of the CLR-transformed data, all nine of these Actinobacteria were identified as significantly more abundant in the tongue from regression of the individual OTUs when using either a 1% false-discovery rate or a Bonferonni correction—these monophyletic taxa all individually show a strong preference for the same body site, and their basal branch was identified as our first phylogenetic factor. The remaining Alpha-, Beta-, Gamma- and Delta-proteobacteria grouped with the Actinobacteria consisted of 31 OTUs, and the Epsilonproteobacteira split from the rest were three unclassified species of the genus *Campylobacter.* The grouping of Actinobacteria with the non-Epsilon Proteobacteria motivates the need for accurate phylogenies in phylofactorization, but also illustrates the promise of identifying clades of interest where the phylogeny is correct and the taxonomy is not.

The second factor ($P = 1.15 \times 10^{-31}$) splits 16 Firmicutes of the class Bacilli from the obligately anaerobic Firmicutes class Clostridia and the remaining paraphyletic group containing Epsilonproteobacteria and the rest. The Bacilli are, on average, $0.3\times$ as abundant in the gut as the paraphyletic remaining OTUs and $3.9\times$ as abundant in the tongue. The 16 Bacilli OTUs factored here contain 12 unclassified species of the genus *Streptococcus,* well known for its association with the mouth (*Guggenheim, 1968*), one member of the genus *Lactococcus*, one unclassified species of the mucosal-associated genus *Gemella*, and two members the family Carnobacteriaceae often associated with fish and meat products (*Leisner et al., 2007*).

The third factor ($P = 1.37 \times 10^{-28}$) separated 15 members of the Bacteroidetes family Prevotellaceae from all other Bacteroidetes and the remaining paraphyletic group of OTUs not split by previous factors. The Prevotellaceae split in the third factor were all of the genus *Prevotella,* including the species *Prevotella melaninogenica* and *Prevotella nanceiensis* found to have abundances $0.3\times$ as abundant in the gut and $4.0\times$ as abundant in the tongue relative to the other taxa from which they were split.

These first three factors capture major blocks visible in the dataset can be used as dimensionality reduction tool with a phylogenetic interpretation (Fig. 4). While traditional ordination-visualization tools may capture larger fractions of variation of the data, phylogenetic factorization yields a few variables—ratios of clades—which capture large blocks of variation in the data and can be traced to single edges in the phylogeny corresponding to meaningful splits between taxa, edges where traits likely arose which
govern the differential abundances across sample sites and environmental gradients or responses to treatments (Fig. 4A, Figs. S4–S8).

Using the KS-test stopping criterion, phylofactorization was terminated at 142 factors, each corresponding to a branch in the phylogenetic tree separating two groups of OTUs based on their differential abundances in the tongue and feces. These 142 factors define 143 groups, or what we call 'bins,' of taxa which remain unsplit by the phylofactorization. The bins vary in size; 112 bins contained only single OTUs, whereas 8 were monophyletic clades and the rest are paraphyletic groups of OTUs, the result of taxa within a monophyletic group being factored, yielding one monophyletic group and one paraphyletic group. Of the 112 single-OTU bins extracted from phylofactorization, 78 were also identified as significant at a false-discovery rate of 1%. Some monophyletic bins included groups of unclassified genera that would not be grouped at the genus level under standard taxonomy-based analyses. For instance, two monophyletic clades of the Firmicutes family Lachnospiraceae were identified as having different preferred body sites, yet both clades were unclassified at the genus level. Taxonomic-based analyses would either omit these unclassified genera, or group them together and make it difficult to observe a signal due to the two sub-groups having different responses to body site.

### Soil microbiome

The soil microbiome dataset was much larger—3,379 OTUs and 580 samples—and a much smaller fraction of the variation could be explained by the dominant factors resulting from phylofactorization. Phylofactorization confirmed that the pH of the environment plays a dominant role in the microbial community composition, consistent with previous analyses based on Mantel tests (Ramirez et al., 2014). Dominance analysis of the generalized linear models associated with each factor determined pH to account for approximately 92.87%, 89.78%, and 92.94% of the explained variance in the first, second, and third factor, respectively. C and N were relatively unimportant, and the dominance of pH in the first three factors can be visualized by ordination-visualization plots of the ILR coordinates of the first three factors (Fig. 5A).

The first factor splits a group of 206 OTUs in two classes of Acidobacteria from all other bacteria: class Acidobacteriia and class DA052 are shown to decrease in relative abundance with increasing pH. The second factor split 31 OTUs in the order Actinomycetales (some from the family Thermomonosporaceae and the rest unclassified at the family level) from the remainder of all other bacteria, and these monophyletic Actinomycetales also decrease in relative abundance with increasing pH. The third factor identified another clade within the phylum Acidobacteria to decrease with pH: 115 bacteria from the classes Solibacteres and TM1.

The fourth factor identifies a monophyletic clade of 193 OTUs in the remainder of phylum Acidobacteria (i.e., those Acidobacteria not mentioned above in factors 1 and 3) as having relative abundances that increase with pH (dominance analysis: 94.79% of explained variance attributable to pH). Unlike the previous three factors above which were acidophiles, this monophyletic group of Acidobacteria consists of alkaliphiles, which

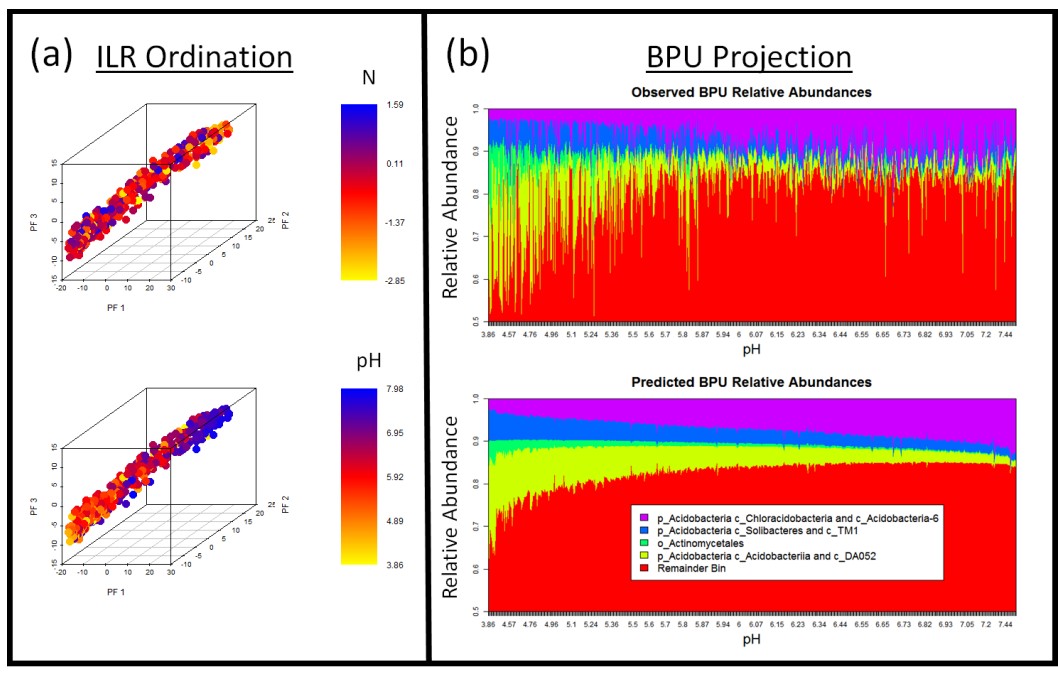

**Figure 5** **Dimensionality reduction and ordination-visualization of soil microbiome dataset.**
Phylofactor presents two complementary methods for projecting and visualizing the high-dimensional
phylogenetically-structured compositional data. (A) The ILR coordinates have asymptotic normality
properties and provide biologically informative ordination-visualization plots. Here, we we see that pH
is a much better predictor than N of the major phylogenetic factors in Central Park soils. Dominance
analysis indicated that pH accounts for approximately 92.87%, 89.78%, and 92.94% of the explained
variance in the first, second, and third factor, respectively, consistent with previous results based on
Bray–Curtis distances and Mantel tests showing the dominance of pH in structuring soil microbiomes
(*Ramirez et al., 2014*). (B) Every edge separates one group of taxa into two, and the disjoint groups of
taxa defined by a common phylofactorization—what we refer to as bins—can be used to amalgamate
OTUs and construct a lower-dimensional, compositional dataset of "binned phylogenetic units" (BPUs).
Plotting the observed relative abundances of BPUs and the relative abundances of BPUs predicted by
phylofactorization yields a simple, phylogenetic story of microbial associations with pH. While low pH
is dominated by Amany Acidobacteria thrive at low pH, there is a monophyletic group of 'acidophobic'
Acidobacteria which thrives at higher pH and isthat includes comprised of classes Chloracidobacteria,
Acidobacteria-6, and S035 and two OTUs unclassified at the class level. The acidophobic Acidobacteria
increase in relative abundance with pH. Also, a monophyletic clade of Actinomycetales is highly
acidophilic and consists of 31 OTUs, 28 of which are unclassified at the family level; the remaining
Actinomycetales are in the remainder bin show a very weak affinity for low pH soils. None of these bins of
OTUs correspond to a single taxonomic grouping, but all of them are characterized and classified by their
locations on one side or the other of four edges within the phylogeny.

includes the classes Acidobacteria-6, Chloracidobacteria, S053 and three OTUs unclassified
at the class level.

The first four factors define 5 bins of OTUs that we refer to as "binned phylogenetic
units" or BPUs: a monophyletic group of Acidobacteria (classes Chloracidobacteria,
Acidobacteria-6, and S035), another monophyletic group of Acidobacteria (classes
Solibacteres and TM1), a monophyletic group of several families of the order
Actinomycetales, a monophyletic group of Acidobacteria (classes Acidobacteriia and
DA0522), and a paraphyletic amalgamation of the remaining taxa. Binning the OTUs

based on these BPUs tells a simplified story of how pH drives microbial community composition (Fig. 5B).

## DISCUSSION

### Overview

We have introduced a simple and generalizable exploratory data analysis algorithm, phylofactorization, to identify clades driving variation in microbiome datasets. Phylofactorization integrates both the compositional and phylogenetic structure of microbiome datasets and produces outputs that contain biological information: effects of independent variables on edges in the phylogeny, including the tips of the tree traditionally analyzed. The output of phylofactorization contains a sequence of "factors," or splits in the tree identifying sub-groups of taxa which respond differently to treatment relative to one-another. The splits identified in phylofactorization need not be splits in the Linnaean taxonomy but can identify strong responses in clades of unclassified taxa. The researcher does not need to choose a taxonomic level at which to perform analysis—those taxonomic levels are output based on whichever clades maximize the objective function, and so researchers will be able to identify multiple taxonomic scales of importance.

Phylofactorization outputs an isometric log-ratio transform of the data with known asymptotic normality properties, coordinates that can be analyzed with standard multivariate methods (*Pawlowsky-Glahn & Buccianti, 2011*). The resulting coordinates correspond to particular edges between clearly identifiable nodes in the tree of life, allowing researchers to annotate a given phylogenetic tree with correlations between clades and various environmental meta-data, sample categories, or experimental treatments.

The phylogenetic inferences obtained by phylofactor would be difficult to obtain with other pre-existing methods. Three iterations of phylofactorizaiton on the oral/fecal dataset yielded the three major splits in the phylogeny, all of which are consistent with known distributions of taxa, none of which would be revealed directly from a taxonomic-based analysis. Algorithms such as phylosignal (*Keck et al., 2016*), which track *P*-values up the tree, yield inferences with nested dependence and identify clades with common significance, yet not necessarily clades with common signal or direction of trend—it is a common signal such as a shared habitat association, not a common significance such as the existence of a habitat association, which better indicates a putative trait driving shared responses in microbes. In the 142 factors above, phylofactor identified numerous clades with common significance yet different signals. Phylogenetic kernel-based methods (*Lozupone & Knight, 2005*; *Purdom, 2011*; *Ning & Beiko, 2015*) use the phylogeny to modify distances/dissimilarities between samples and can easily differentiate between oral and fecal body sites, but such nominally similar methods classify and differentiate sites, which is a conceptually different from classifying and predicting OTU abundances given their place in the phylogeny. Similarly, phylogenetic comparative methods (PCMs) (*Felsenstein, 1985*; *Harvey & Pagel, 1991*) such as phylogenetic generalized least squares (*Grafen, 1989*; *Martins & Hansen, 1997*) are nominally similar but conceptually distinct from phylofactor; while PCMs aim correct for the dependence of observations of evolutionarily related taxa, our

goal is not to correct for such dependence but to infer precisely where on the tree such dependence appears to arise. While many tools exist for the nominally similar task of "using the phylogeny to analyze microbiome data", the utility of phylofactorization lies in its ability to construct variables corresponding to edges in the phylogeny along which putative functional ecological traits may have arisen, constructed so as to avoid nested dependence and overlapping comparisons that frequent the analysis of hierarchically structured, and constructed in a manner appropriate for compositional data.

## Future work

The generality of phylofactorization opens the door to future work employing phylofactorization with other objective functions. As we showed with the human oral/fecal microbiomes, phylofactorization is not restricted to basal clades, but includes the tips as possible clades of interest, but the objective function we used minimized residual variance in the whole community and thereby may prioritize deeply rooted edges or abundant taxa with weaker effects over individual OTUs with stronger effects. Other objective functions could be constructed to meet the needs of the researcher. If researchers are interested in identifying basal lineages, their objective function can weight edges based on distance from the tips. A researcher interested in fine-tuning the evolutionary assumptions in phylofactorization can define an objective function that increases explicitly with the length of the edge being considered to reflect an assumption that the probability of a trait arising increases with the amount of time elapsed.

Each edge identified in phylofactorization corresponds to two bins of taxa on each side of the edge, and consequently phylofactorization brings in two complementary perspectives for analyzing the data: factor-based analysis and bin-based analysis. Factor-based analysis looks at the each factor as an inference on an edge in the phylogeny, conditioned on the previous inferences already made, and indicating that taxa on one side of an edge respond differently to the independent variable compared to taxa on the other side of the edge. Bin-based analysis, on the other hand, looks at the set of clades resulting from a certain number of factors—what we call a "binned phylogenetic unit" (BPU). These bins will create a lower-dimensional, compositional dataset and can be freed from the underlying ILR coordinates for different analyses on these amalgamated clades. BPU-based analysis can inform sequence binning in future research aimed at controlling for previously-identified phylogenetic causes of variation, and combine the effects of multiple up-stream factors for predictions of OTU abundance. See Supplemental Information 1 for a more detailed discussion of factor-based and bin-based analyses.

One conceptual challenge with phylofactorization cross-validation and BPU-based analysis is the treatment of paraphyletic "remainder" clades which occur after factorization of edges within groups that split a monophyletic clade from a "remainder," paraphyletic clade or BPU. These paraphyletic BPUs are informative as they are the necessary contrasts with the monophyletic group, conditioned on previously identified phylogenetic factors, and thus serve as a means of identifying the presence or absence of a trait in the monophyletic group. For instance, phylofactorization of vertebrates based on their relative abundance in the air should identify two monophyletic groups: birds and bats. The

monophyletic groups are defined by the presence of a specialized adaptation—wings—and the paraphyletic remainder is defined by the absence of wings; any future treatment that differentially affects winged vs. non-winged organisms will differentially affect the abundances of these monophyletic groups relative to the paraphyletic remainder. Conversely, "legs" in the order Squamata disappeared along edges corresponding to legless lizards and snakes; in this case, the monophyletic clades are defined by the absence of a specialized adaptation, legs, and the paraphyletic remainder is defined by the presence of legs. Whether a trait can be flexibly defined as either the presence of absence of a specialized adaptation is an important theoretical consideration, but considering monophyletic BPUs and their complementary paraphyletic BPUs as equally likely to have the presence or absence of a trait and as the necessary comparison for cross-validation is an important empirical consideration. Phylofactorization pinpoints the contrasts to be made for microbial genomic and physiological studies aiming to identify the causes of differential abundances or responses to treatments across OTUs. Genomic/physiological investigations and cross-validation of phylofactorization must understand and grapple with the choice of appropriate comparisons of taxa, and that may require a fair contrast of a monophyletic clade to a paraphyletic one.

Phylofactorization will benefit from community discussion and further research overcoming general statistical challenges common to greedy algorithms and analysis of phylogenetically-structured compositional data. For instance, the log-ratio transform at the heart of phylofactorization requires researchers deal with zeros in compositional datasets. While there are many methods for dealing with zeros (*Aitchison, 1986*; *Martín-Fernández, Barceló-Vidal & Pawlowsky-Glahn, 2003*; *Pawlowsky-Glahn & Buccianti, 2011*), it's unclear which method is most robust for downstream phylofactorization of sparse OTU tables. Second, phylofactorization as presented here only sequentially infers ILR elements and does not allow for simultaneous inference of ILR basis elements—the set of factors identified after $n$ iterations may explain less variation combined than an alternative set of factors that did not maximize the explained variance at each iteration. This limitation may be overcome by running many replicates of a stochastic greedy algorithm and choosing that which maximizes the explained variance after $n$ factors. Third, the researcher must choose an objective function which matches her question, and future research can map out which objective functions are appropriate for which questions in microbial ecology. Fourth, like any method performing inference based on phylogenetic structure, phylofactorization assumes an accurate phylogeny. Accurate statistical statements about a researcher's confidence in phylofactors must incorporate the uncertainty in our constructed phylogeny. Fifth, phylogenetic-based methods may be sensitive to the binning and filtering methods for sequence-count data. Our filtering methods here were chosen to allow somewhat standard and simplified analysis of common OTUs present in many of the samples, but these filtering protocols might not be optimal for researchers with other research questions and objective functions. Future work can investigate the sensitivity of phylofactorization to myriad binning and filtering methods given the objectives and objective function of the researcher. Finally, future research can investigate the unique kinds of errors in phylofactorization: in addition to the multiple-hypothesis testing of

edges, phylofactorization may propagate errors in the greedy algorithm, and, even when taxa are correctly factored into the appropriate functional bins, the presence of multiple factors in the same region of the tree can lead to uncertainty about the exact edge along which a putative trait arose (see Supplemental Information 1 for more discussion on the uncertainty of which edge to annotate).

Incorporating that phylogenetic structure into the analysis of microbiome datasets has been a major challenge (*Martiny et al., 2015*), and now phylofactorization provides a general framework for rigorous exploration of phylogenetically-structured compositional datasets. The soil dataset analyzed above, for instance, contains 3,379 OTUs and 580 samples, and phylofactorization of the clades affected by pH in the soil dataset yielded not just the three dominant factors used for ordination-visualization, but 2,430 factors in all, each with an intricate phylogenetic story. Many Acidobacteria are acidophiles, but some—Chloracidobacteria, Acidobacteria-6, S035, and some undescribed classes of bacteria factored here—appear to be alkaliphiles. By incorporating the phylogenetic structure of microbiome datasets, the big data of the modern sequence-count boom just got bigger, and future research will need to consider how to organize, analyze and visualize the large amounts of phylogenetic detail that can now be obtained from the analysis of microbiome datasets.

## CONCLUSIONS

Phylofactorization is a robust tool for analyzing marker gene sequence-count datasets for phylogenetic patterns underlying microbial community responses to independent variables. Phylofactorization accounts for the compositional nature of the data and the underlying phylogeny and produces inferences that are independent and more powerful than application of the ILR transform to the rooted phylogeny. The R package 'phylofactor' has built-in parallelization that can be used to analyze large microbiome datasets, and allows generalized linear modeling to identify clades which respond to treatments or multiple environmental gradients.

Phylofactorization can connect the pipeline of microbiome studies to focused studies of microbial physiology. As researchers identify lineages with putative functional ecological responses, taxa within those lineages—even if they are not the same OTUs—can be cultivated and their genomes screened to uncover the physiological mechanisms underlying the lineages' shared response.

Phylofactorization improves the pipeline for analyzing microbiome datasets by allowing researchers to objectively determine the appropriate phylogenetic scales for analyzing microbiome datasets—a family here, an unclassified split there—instead of performing multiple comparisons at each taxonomic level. Instead of principle components analysis or principle coordinates analysis, phylofactorization can be used as for exploratory data analysis and dimensionality reduction tool in which the ''components'' are identifiable clades in the tree of life, a far more intuitive and informative component for biological variation than multi-species loadings.

Phylofactorization can allow researchers to annotate online databases of the microbial tree of life, permitting predictions about the physiology of unclassified and uncharacterized

life forms based on previous phylogenetic inferences in sequence-count data. By allowing researchers to make inferences on the same tree and potentially annotate an online tree of life, phylofactorization may bring on a new era of characterizing high-throughput phylogenetic annotations, filling in the gaps the microbial tree of life.

An R package for phylofactorization with user-friendly parallelization is now available online at https://github.com/reptalex/phylofactor.

## ACKNOWLEDGEMENTS

ADW would like to acknowledge L Ma for his feedback and help incorporating this method into the statistical literature. This paper is published by support from and in loving memory of D Nemergut.

### Funding

This work was supported by startup funds for Diana Nemergut, provided by the Duke University Department of Biology. There was no additional external funding received for this study. The funders had no role in study design, data collection and analysis, decision to publish, or preparation of the manuscript.

### Grant Disclosures

The following grant information was disclosed by the authors:
Duke University Department of Biology.

### Competing Interests

The authors declare there are no competing interests.

### Author Contributions

- Alex D. Washburne conceived and designed the experiments, performed the experiments, analyzed the data, contributed reagents/materials/analysis tools, wrote the paper, prepared figures and/or tables, reviewed drafts of the paper.
- Justin D. Silverman, Jonathan W. Leff, Dominic J. Bennett, John L. Darcy and Sayan Mukherjee contributed reagents/materials/analysis tools, reviewed drafts of the paper.
- Noah Fierer contributed reagents/materials/analysis tools, reviewed drafts of the paper, logistical management and support for collaboration across the country.
- Lawrence A. David reviewed drafts of the paper, logistical management and support for collaboration across the country.

### Data Availability

GitHub: https://github.com/reptalex/phylofactor.

### Supplemental Information

Supplemental information for this article can be found online at http://dx.doi.org/10.7717/peerj.2969#supplemental-information.

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
