# Peer review of "Phylogenetic factorization of compositional data yields lineage-level associations in microbiome datasets"

_PeerJ, doi:10.7717/peerj.2969_

## Round 0.1 · original submission · Major Revisions

Both reviewers find important merits in the work, but they also raise several concerns about the presentation of the results, the assessment of the new method, and the software availability. Please address these concerns in the revised version.

·

Basic reporting

No Comments

Experimental design

No Comments

Validity of the findings

The study by Washburne et al. presents a novel approach for marker gene-based community analysis which focuses on edges in the phylogeny as opposed to single (or multiple) pre-defined taxonomic or phylogenetic levels.
As I am not an expert in the mathematical approaches employed I cannot verify the validity of the methods employed. Hence I suggest that the editor looks for such verification elsewhere. Nevertheless I could indeed follow their rationale and find the work novel, innovative and timely.
The simulations employed mimicked scenarios with numbers of OTUs, affected clades and samples quite reduced when compared to known natural scenarios. Also, the level of OTU filtering carried out with the oral and soil datasets was quite severe. Finally, the use of reference-based clustering with these datasets, as well as employing the actual Greengenes reference tree instead of one derived from the sequences itself poses some questions with regards to the efficacy of the developed method with the current datasets being generated. Nevertheless, I believe that such shortcomings do not diminish the importance of the results contained within the MS, as well as the described method. Moreover, the authors acknowledge throughout the manuscript that the method is still in its infancy, and call for a community-level discussion on the possible issues associated, most of which are correctly identified in the MS. Finally, the availability of a dedicated R package will be most welcomed by the community.

Additional comments

If I followed correctly, phylofactorization always yields one BPU composed of ‘remains’ paraphyletic clades. Is that so? As there are no particular common traits/common phylogenetic history for members of this BPU, maybe it would be best if the authors identify this group with a different terminology (not BPU). In this regard, BPUs are phylogenetically consistent groups with a coordinated response against the external variable(s) being analyzed, while the ‘remains’ group(BPU) does not show such response, nor is it phylogenetically consistent.

There is also another issue which is not clear to me; once a particular edge has been pinpointed by the method it seems that sub-clades within it will not be analyzed. Is that so because I find several sentences apparently contradictory in this regard? I guess that if one of such sub-clades had a stronger effect it could have been pinpointed in an earlier iteration, and then kin clades could eventually also be pinpointed. The authors also comment that this issue will also be heavily dependent on the objective function employed. Is this interpretation correct? Could the authors comment on it (and possibly improve their explanation of this issue in the MS)?

What follows are minor comments on the MS which may or may not help the authors improve the MS. The authors should by no means feel required to address them.
Minor:
-I´m not 100% for the term phylofactorization, how about something easier like phylofinder (or any other similar term). I´m not sure if the moiety ‘factorization’ stems from the mathematics being employed, but I feel that for most users the term will be confusing, if not misleading. First, it feels odd to identify edges/clades as ‘factors’. Second, one would assume that ‘factor’ in this context is the external variable being assessed.

-Similarly, ‘dimensionality-reducing tool’ while clearly accurate is quite reminiscent of pca,pco, eigen factor decomposition and the like, and seems somehow counterintuitive. Furthermore, is it really an ordination tool? I confess that I am unsure as to what ‘ordination’ means, other than ordination in reduced space (linked to the issue above).

Minor specific comments:
L11-12: It seems to me that ‘OTU’ is not appropriate here, is it? I guess that the correct term could be BPUs, yet this term cannot be used as it has not yet been introduced, how about ‘partitions in the phylogenetic space’ or something in that line?
L16: needs references.
L36: ‘allowing dimensionality reduction’. Similar to my previous comment; I feel that the term ‘dimensionality reduction’ overcomplicates the statement.
L39: Maybe this sentence could go better before “In this paper”
L59: if each factor corresponds with an edge; couldn´t you use ‘edge’ directly to reduce the number of terms involved in the explanation?
L74-78: This sentence is not easy to follow, afterwards everything makes sense however it could be better if sentences were not as dense this early in the MS.
L68-89: This introductory paragraph seems too long, consider summarizing a bit more the results at this point.
L182: “for the annotation of online trees of life”; this seems a little too far in the future (mid-term). I accept that that is a desirable and foreseeable scenario. However, I would suggest changing the sentence to something like ‘annotation of phylogenetic trees’.
L293: Which value? Why?
L360-379: It feels odd to find this summary of results at the beginning of the results section. Also, isn´t it kind of a repetition of paragraph L68-89?
L395-396: This is a little of an overstatement; one can assume correlation between phylogeny and ecological/physiological coherence. However, I feel that ‘a clear biological interpretation’ is too much to say.
L481, and in general in the MS. The authors should discuss briefly on previous similar methods such as phylosignal, cDPCoA or unifrac (I seem to remember that either the unifrac suite or Qiime allowed the detection of the particular clades driving the differences in the dataset), if only to ascertain their proposed method superiority. For instance cDPCoA also identified Acidobacteria relative abundance as correlated with soil pH. However, it is unable to identify sub-groups within such clade showing opposing behaviors (as shown in the MS)
L484-485-It may not be clear to the reader the difference between signal and significance.
L489-491. This sentence could be removed. The MS is already quite long, which makes sense as it introduces many novel concepts etc. However, the info within the sentence has been mentioned often previously.(same with L68-89 + L360-379)
L557-560. I´m not sure if I follow this correctly; weighting each edge alone or each edge plus all edge distances until root? Also, the whole point and rationale behind phylo-oriented analysis is the concept that traits/ecophysiology conservation is correlated with phylogenetic relatedness. Hence, with these analyses one is always ‘interested in identifying putative traits’.
L567-569. I understand the use of the term BPU. However, the proposed procedure should always use discrete monophyletic clades plus a ‘residual ‘remains’ paraphyletic bin. Why not call them phylogenetic units and then identifying the ‘remainder’ paraphyletic BPU with a specific name. This could be useful as I believe that phylogenetic units will in the near future be the target of novel approaches etc and it would be useful to use common terminology.
L572-574 This sentence is a little odd. Also it seems that these concepts are repeated too much along the MS.
L627. Wasn´t it mentioned before that right now the technique does not allow for the analysis of the effect of multiple factors on the dataset (e.g. sex + diet), or maybe not the effect of such factors on the same bin? I gather that if it allows the analysis of multiple gradients it should permit multiple factors. This should be better explained.
* * *
Overall figure legends could be improved and explain a bit more; for instance in figure 4b; what are the taxonomic assignments depicted? the number of colored tips in each group does not correspond to the number of taxonomic assignment for that group, is it a consensus taxonomy?. Also, in figure 5 b, what is the difference between both charts? Which one is the one referred in the text (and what is the other)? What is the meaning of ‘predicted BPUs’ in the bottom chart?

SI:: Green figure: section ‘d’ could be better explained. Also, as there is no panel ‘d’ this discussion could be integrated in ‘c’.
SI:: “”Thus, phylofactorization as described
here is a recommended method when researchers anticipate there are a small
number of clades with meaningful efects in an experiment, and should be used
with caution when the number of traits driving variation in the response variable
is larger than the number of taxa.””
 Could you explain this a bit further as it seems important. Also what do you mean by taxa here, tips of the tree?
SI: “Compositional benchmarking”; In order to reduce computation effort, couldn´t the tip-most edges be collapsed somehow? This could be a good strategy to explore (I imagine that in actual real life scenarios the phylogenetic trees will be pretty large and the tipmost edges will not likely be pinpointed by the method).
SI:: Figure 8 is quite intuitive, consider moving it to the main MS. Also, was this kind of tree representation chosen for a particular reason?

Reviewer 2 ·

Basic reporting

The article is very clear, well-written, and most of the literature/references are sufficient, with the exception of missing some fundamental references (see below). The figures and tables are proper and readable and the raw data has been made available.

Improvements: The author should cite original literature about the comparative method and independent contrasts, which provide the underlying theory behind the method. Two refs to include:

Felsenstein, Joseph (January 1985). "Phylogenies and the Comparative Method". The American Naturalist. 125 (1): 1. doi:10.1086/284325.

Harvey, Paul H.; Pagel, Mark D. (1991). The Comparative Method in Evolutionary Biology. Oxford: Oxford University Press. p. 248. ISBN 9780198546405.

It also seems appropriate to have original literature about self-balancing binary tree algorithms.

Figure 1 - does not have (A) and (B) labels.

Experimental design

This is original research that is well-defined and meaningful, and adding an explicit phylogenetic approach to this problem is very welcome.

The investigation is rigorous and the procedure explained well (though I would have appreciated a cartoon example of how the method works sort of like the original UniFrac paper).

The methods should be easily replicated, though I found it onerous since the code is in github and I did not find any kind of tutorial on how to use all the various scripts.

The best way to use new R code is if it can be accessed via CRAN (in the CRAN depository). Like Vegan. This would greatly increase it's user friendliness. Never having created a CRAN library, I do not know how hard this is.

Failing this, the next best thing is to have a really good tutorial. I did not find this in any of the documentation. Combining that with good simple test files would have helped me test out and verify the code. It would also insure that researchers actually use the software.

Validity of the findings

I do have some concerns about the findings and the conclusions. The authors should really address these issues at least to the reviewers satisfaction.

I'm generally concerned about two things: (1) The approach as the authors say maximizes the chance of finding differences or finding clades that differ among treatments. This is ok, as a tool, but I imagine we will find something if we look hard enough.

What about splitting the data and having a test set and a exper set? Like a training set or a control. The control would find clade x and y. Would the test set(s) repeat this?

(2) The clades discovered seem pretty standard. I think they would be found easily with non-phylogenetic methods (taxonomic). Since so few were found to be different in both examples and they seem they would be different with other methods, what have we gained here? The resolution and number of results returned seems poor.

Figure 4: This comparison is really unconvincing. The tongue vs. the feces? They are so very different, it is surprising that you only identified 3 clades. Is this really much different than you'd find with other methods? There are loads of taxa different in oral than feces and one would presume that many clades t many levels would reflect this. Unless I'm missing something, this is not a very convincing test of utility for new discovery.

Figure 5: Again, I'm not convinced that phylofactorization is telling us a lot new here that other approaches haven't. Ph is critical in soils (well known) and the basic taxonomic appraoches tell us this. What is really unique here?

---

## Round 0.2 · accepted · Accept

The revised version satisfactorily addressed all of the reviewers's concerns

·

Basic reporting

no comment

Experimental design

no comment

Validity of the findings

no comment

Additional comments

no comment

Reviewer 2 ·

Basic reporting

The authors addressed my concerns.

Experimental design

The authors addressed my concerns.

Validity of the findings

The authors addressed my concerns.